# Asymmetric transformation of achiral gold nanoclusters with negative nonlinear dependence between chiroptical activity and enantiomeric excess

Chang Liu[1,2], Yan Zhao[1,2], Tai-Song Zhang[1], Cheng-Bo Tao[1], Wenwen Fei[1], Sheng Zhang [1] & Man-Bo Li [1] ✉

The investigation of chirality at the nanoscale is important to bridge the gap between molecular and macroscopic chirality. Atomically precise metal nanoclusters provide an ideal platform for this research, while their enantio-pure preparation poses a challenge. Here, we describe an efficient approach to enantiopure metal nanoclusters via asymmetric transformation, that is, achiral $Au_{23}(SC_6H_{11})_{16}$ nanoclusters are converted into chiral and enantiopure $Au_{24}(L)_2(SC_6H_{11})_{16}$ nanoclusters by a chiral inducer phosphoramidite (L). Two enantiomers of $Au_{24}(L)_2(SC_6H_{11})_{16}$ are obtained and the crystal structures reveal their hierarchical chirality, which originates from the two introduced chiral L molecules, the transformation-triggered asymmetric rearrangement of the staple motifs on the surface of the gold core, and the helical arrangement of nanocluster molecules. The construction of this type of enantio-merically pure nanoclusters is achieved based on the easy-to-synthesize and modular L. Lastly, the chirality-related chiroptical performance was investigated, revealing a negative nonlinear CD-ee dependence.

Chirality is a fundamental characteristic in nature and can be found at various scales, from small organic molecules to macroscopic materials[1–4]. The same component materials with different chiroptical activities show distinct performance[5]. As a typical example, the S and R enantiomers of organic molecules with asymmetric $sp^3$ carbons display different even completely opposite biological activities[6]. That is why the asymmetric synthesis is so important in medicinal chemistry. Based on the analysis of small organic molecules' structures, we can clarify the origin of their chirality and achieve efficient enantiomer synthesis. However, when the scale of materials comes to nanometers or even micrometers (e.g., organic ligand-protected metal nano-particles), things become complex[7–10]. The chirality of nanoparticles can originate from the chiral organic ligands, the asymmetric arrangements of metal atoms or metal-ligand motifs, as well as the

particles' asymmetric packing at a higher scale[11,12]. The complex chir-ality and the undefined structure of nanoparticles make the determi-nation of chirality origin, the achievement of enantiopure isomers, and the investigation of the related performance challenging.

The atomically precise metal nanocluster constitutes an ideal platform for the research of complex systems' chirality at atomic level[13]. Moreover, chiral metal nanoclusters show unique properties and potential applications in catalysis, sensing, as well as biomedicine[14,15]. Thus, the investigation of the chirality of metal nanoclusters has received tremendous attention[16–20]. In spite of the significant progress, the achievement of enantiopure metal nanoclus-ters is desired but remains a great challenge. Based on the previous reports, some methods were developed for acquiring metal nanoclusters with enantiomeric excess. These methods include: (1)

[1]Institutes of Physical Science and Information Technology, Key Laboratory of Structure and Functional Regulation of Hybrid Materials of Ministry of Education, Anhui University, 230601 Hefei, P. R. China. [2]These authors contributed equally: Chang Liu, Yan Zhao. ✉e-mail: mbli@ahu.edu.cn

enantioseparation of racemic mixtures of metal nanoclusters by chiral high-performance liquid chromatography (HPLC)[21–24] or resolving agents;[25–27] (2) enantioselective phase transfer of chiral nanoclusters;[28,29] (3) direct synthesis of enantiopure nanoclusters by using chiral organic ligands as the precursors[30–37]. However, the method for transforming an achiral metal nanocluster into a chiral one is limited. There are a few reports on the chiral ligand exchange of achiral metal nanoclusters[38,39], in which the chirality generally comes from the chiral ligands on the surface. In view of hundreds of achiral metal nanoclusters being reported[13] and the wide application of asymmetric organic transformations in pharmaceutical synthesis and industry chemistry[40,41], we envisioned the possibility of using a chiral inducer to realize the asymmetric transformation of achiral metal nanoclusters. This strategy would be developed into a general protocol for the construction of enantiopure metal nanoclusters.

We have a long-standing interest in the chirality transfer of organic molecules[42–45] as well as the construction of functional metal nanoclusters[46,47]. We developed a surface phosphorization method[48] and realized the structural evolution of thiolate-protected gold nanoclusters, which has been proven to be effective to modify the nanoclusters' structure and performance[48,49]. Inspired by these achievements, in this work, we realized the asymmetric transformation of an achiral gold nanocluster by using an easily synthesized and modular phosphine inducer. The transformation is enantiodivergent and the two optical pure enantiomers of the as-transformed nanoclusters can be obtained respectively. More interestingly, this strategy not only introduces the chirality of the phosphine inducer but also triggers the asymmetric arrangement of -S-Au-S- and -S-Au-S-Au-S- staple motifs that were originally symmetric on the surface of the gold nanocluster. An intriguing helical arrangement of nanocluster molecules was also discovered. Thus, the asymmetric transformation leads to a hierarchically chiral structure of the gold nanocluster.

## Results

### Asymmetric transformation process

A facilely synthesized $Au_{23}(SC_6H_{11})_{16}$ (abbreviated as $Au_{23}$) was used as the model nanocluster for the investigation. It is an achiral gold nanocluster, bearing an $Au_{13}$ kernel with symmetrically arranged two -S-Au-S-Au-S-Au-S- and four -S-Au-S- staple motifs on the surface[50]. Meanwhile, it is a representative thiolate-protected gold nanocluster, which has been applied as a typical example for the studies of

structural anatomy, metal doping, and structure-property correlation[51–53]. We initiated our studies by screening different kinds of chiral phosphine ligands to react with $Au_{23}$ (Supplementary Table 1). Butane-2,3-diybis(diphenylphosphine) with two chiral sp$^3$ carbons was reactive to $Au_{23}$ but showed poor selectivity and resulted in wide product distribution (Supplementary Fig. 1a). Axially chiral 2,2'-bis(diphenylphosphino)−1,1'-binaphthyl (BINAP) with sterically hindered phosphine sites displayed low activity to $Au_{23}$ instead, and led to the recovery of the starting materials (Supplementary Fig. 1b). Chiral phosphoric acid resulted in the decomposition of $Au_{23}$, probably due to its strong acidity. Delightedly, a "privileged" ligand phosphoramidite[48] was found to demonstrate satisfactory reactivity as well as good selectivity (Supplementary Fig. 2). This ligand was first introduced by Feringa and coworkers[54,55] and Alexakis and coworkers[56] in the late 1990s for copper-catalyzed asymmetric conjugate additions. They are modular, easy to synthesize, and widely applied as a versatile and readily accessible class of chiral ligands in asymmetric catalysis[56]. Racemic phosphoramidite with a diethylamine module ($Rac$)-$L_1$ was prepared initially to react with $Au_{23}$ at room temperature (Fig. 1a). This reaction is efficient, giving an exclusive nanocluster product in high yield. The obtained nanocluster showed a different polarity to $Au_{23}$ based on the preparative thin layer chromatography (PTLC), demonstrating that a different nanocluster ($LC_1$) was formed (Supplementary Fig. 2). Meanwhile, the ultraviolet and visible (UV-vis) spectrum of $LC_1$ is quite similar to that of $Au_{23}$, and only a slight blueshift of the characteristic peaks at 460 and 575 nm was observed during the transformation (Fig. 1b). This result suggests that the structure of $LC_1$ would be relevant to that of $Au_{23}$. ($R$)-$L_1$ and ($S$)-$L_1$ reacted with $Au_{23}$, affording ($R$)-$LC_1$ and ($S$)-$LC_1$ with the same UV-vis spectra (Fig. 1b). Electrospray ionization mass spectrometry (ESI-MS) was used to determine the composition of the as-obtained nanocluster. A single peak at $m/z = 3806.15$ was observed on the positive mode with the addition of CsOAc. This peak is assigned to $[Au_{24}(L_1)_2(SC_6H_{11})_{16} + 2Cs]^{2+}$ (Fig. 1c), and the experimental isotope patterns match very well with the calculated one (Fig. 1c, inset). The ESI-MS result suggests that the as-obtained nanocluster is charge-neutral and determines the molecular formula of $LC_1$ to be $Au_{24}(L_1)_2(SC_6H_{11})_{16}$.

### Crystal structures analysis

Single crystals of ($R$)-$LC_1$ and ($S$)-$LC_1$ were obtained in the mixed solvents of dichloromethane and hexane. From the crystal

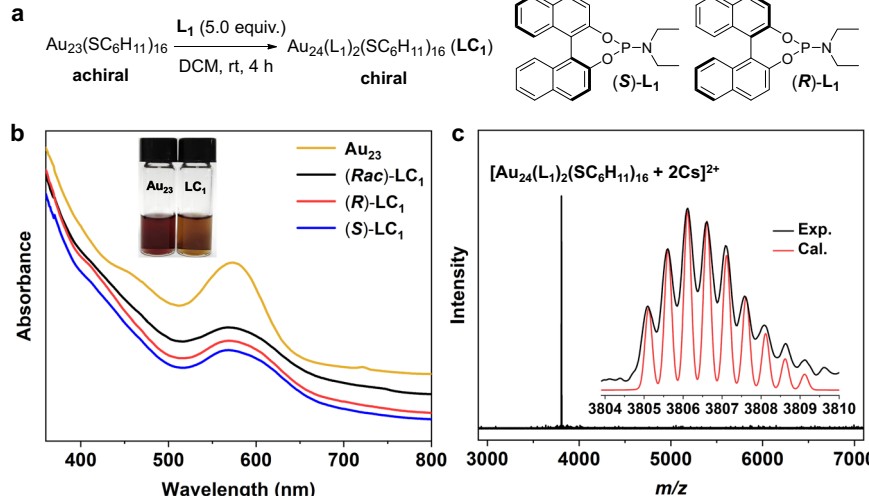

**Fig. 1 | Procedure and characterizations of asymmetric transformation.**
**a** Procedure of the asymmetric transformation. **b** UV-vis spectra of $Au_{23}$ (yellow trace), ($Rac$)-$LC_1$ (black trace), ($R$)-$LC_1$ (red trace) and ($S$)-$LC_1$ (blue trace). Inset: the

photographs of $Au_{23}$ and $LC_1$ in DCM. **c** ESI-MS spectrum of $LC_1$. Inset: the experimental (black trace) and calculated (red trace) isotope patterns. Source data are provided as a Source Data file.

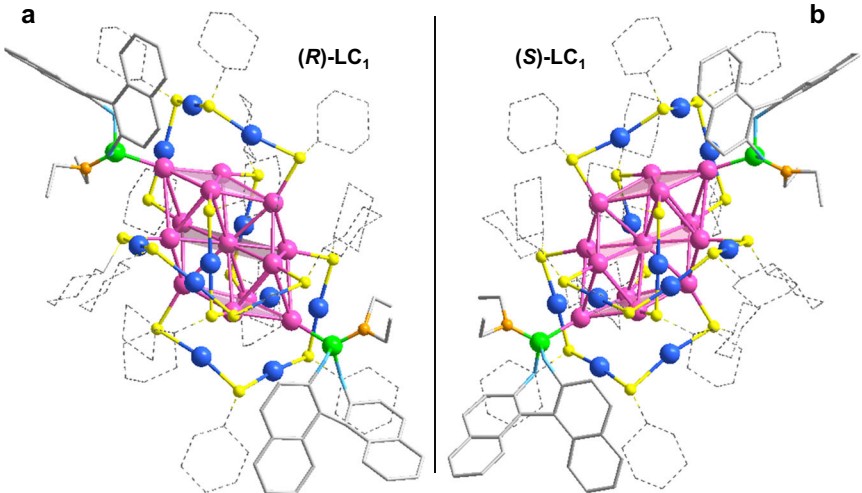

**Fig. 2 | Crystal structures. a** Structure of (**R**)-**LC₁**. **b** Structure of (**S**)-**LC₁**. Color label: Au = pink, blue; S = yellow; P = green; N = orange; O = light blue; C = gray. H atoms are omitted for clarity.

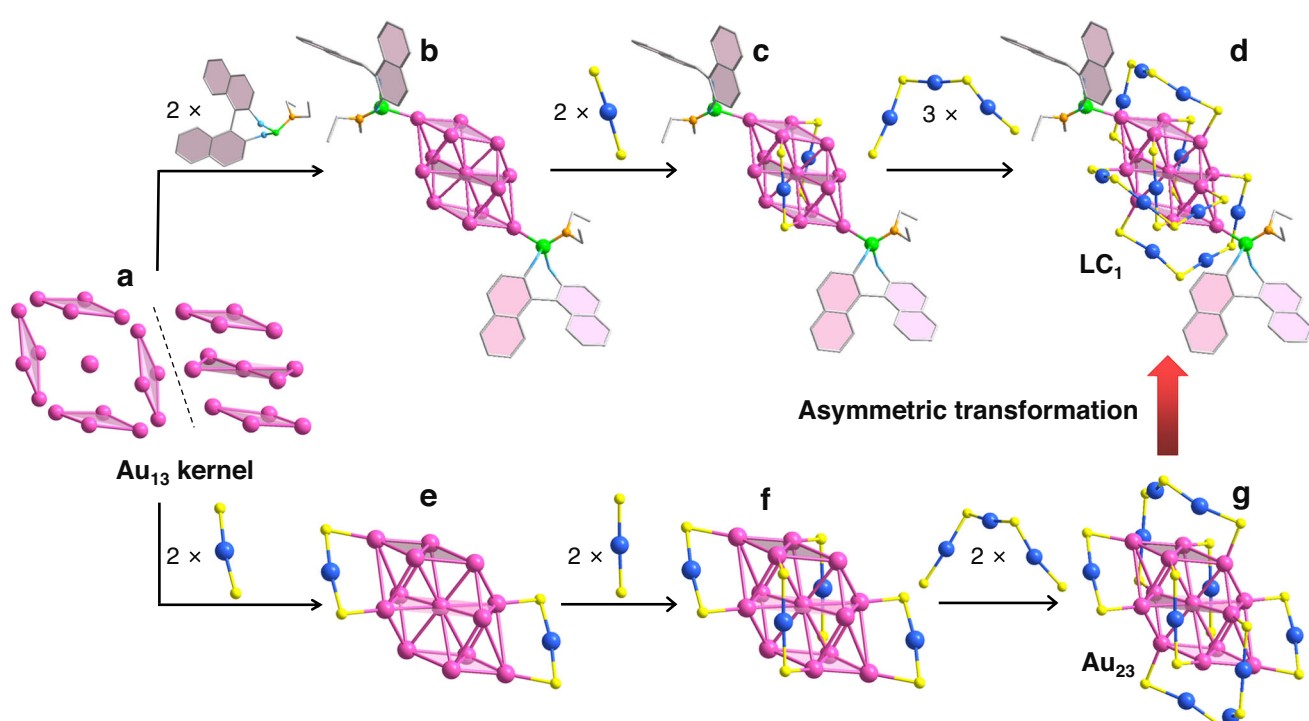

**Fig. 3 | Structural anatomy of LC₁ and Au₂₃. a** The Au₁₃ kernel. **b** Au₁₃ kernel with two phosphoramidites **L₁**. **c** Au₁₃ kernel with two phosphoramidites **L₁** and two -S-Au-S- motifs. **d** The total structure of **LC₁**. **e** Au₁₃ kernel with two -S-Au-S- motifs. **f** Au₁₃ kernel with four -S-Au-S- motifs. **g** The total structure of Au₂₃. Color label: Au = pink, blue; S = yellow; P = green; N = orange; O = light blue; C = gray. H atoms are omitted for clarity.

structures revealed by single-crystal X-ray diffraction (SCXRD), (**R**)-**LC₁** and (**S**)-**LC₁** all crystallized in the enantiomorphic P2₁ space group, which is different from the centrosymmetric Ccca space group of Au₂₃[44]. The (**R**)-**LC₁** and (**S**)-**LC₁** nanocluster molecules take a similar 'ABCD' stacking sequence along the [100], [010] and [001] directions based on the observation of their crystallographic arrangements (Supplementary Figs. 4 and 5). The total structures of (**R**)-**LC₁** and (**S**)-**LC₁** demonstrate that the as-transformed nanoclusters are composed of 24 gold atoms, 16 cyclohexanethiols, and two phosphoramidites **L₁** (Fig. 2), which is consistent with the molecular formula revealed by ESI-MS. (**R**)-**LC₁** and (**S**)-**LC₁** cannot overlap with each other, and show near-perfect mirror images (Fig. 2), indicating

that the (**R**)- and (**S**)-phosphoramidite-induced transformations of achiral Au₂₃ leads to a pair of enantiomeric nanoclusters.

One of the enantiomers (**R**)-**LC₁** was selected as an example for further structural anatomy. As shown in Fig. 3, **LC₁** consists of an Au₁₃ kernel, which can be seen as an Au₄-Au₅-Au₄ sandwich structure, and also a central gold atom surrounded by four common-vertex rhombic blocks (Fig. 3a). The central gold atom is associated with the sur-rounded ten gold atoms with the average Au–Au bond of 2.84 Å (Supplementary Fig. 6) except for the two gold atoms on the vertices (which are far away from the central gold atom with the distances of 4.28 and 4.30 Å). The two vertices are coordinated with the phosphine sites of phosphoramidites **L₁** (Fig. 3b), and the other ten gold atoms are

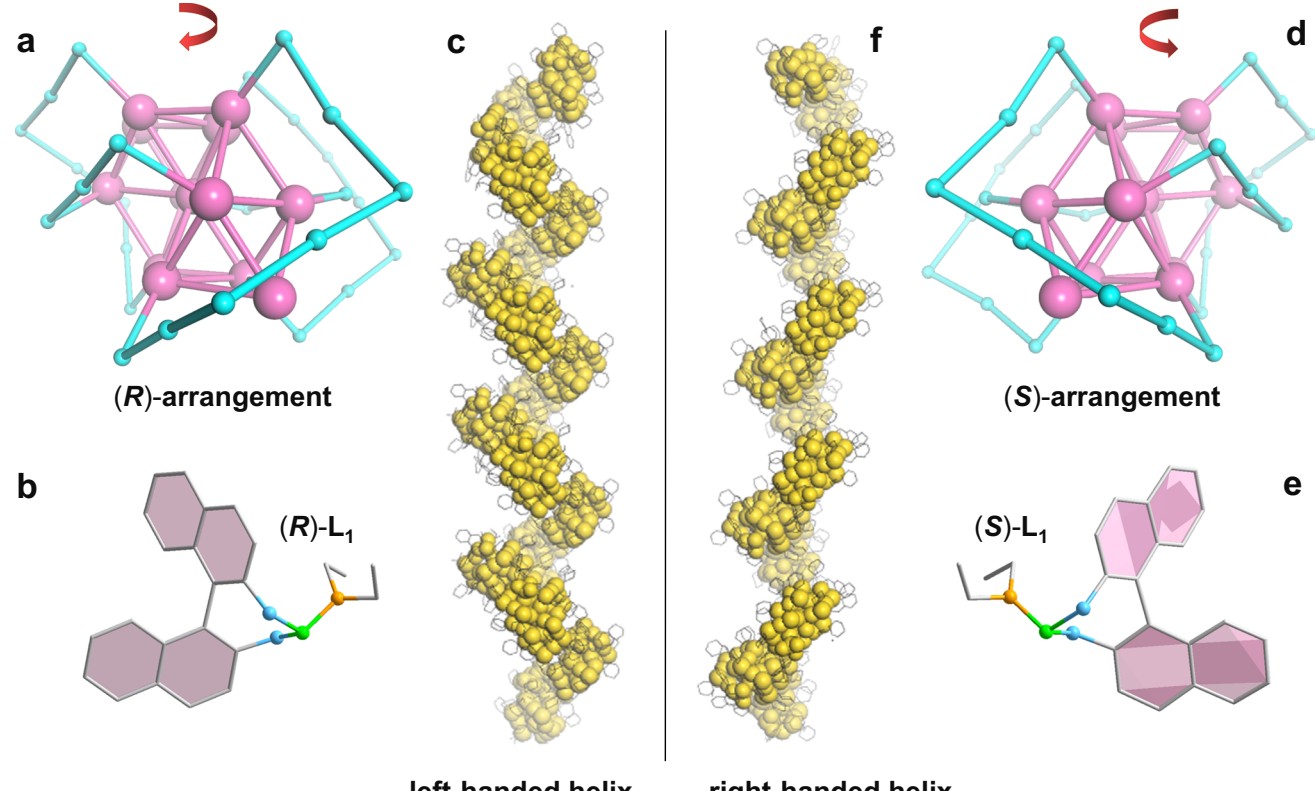

**Fig. 4 | Representation of hierarchically chiral structures of (*R*)-LC₁ and (*S*)-LC₁.** **a**, **d** The asymmetric arrangement of -S-Au-S- and -S-Au-S-Au-S-Au-S- motifs on the surface of the Au₁₃ kernel. **b**, **e** The intrinsic chirality of phosphoramidite ligand at the two vertices of the Au₁₃ kernel. **c**, **f** The helical arrangement of nanocluster molecules. Color label: Au₁₃ kernel = rose red; -S-Au-S- and -S-Au-S-Au-S-Au-S- staple motifs = cyan; Helical nanocluster molecules = yellow; P = green; N = orange; O = light blue; C = gray. H atoms are omitted for clarity.

covered by two -S-Au-S- and three -S-Au-S-Au-S-Au-S- staple motifs (Fig. 3c, d). Notably, **LC₁** and Au₂₃ nanoclusters have similar sandwich Au₁₃ kernels. A careful comparison of the Au–Au distances on the upper and lower Au₄ layers reveals that the Au₁₃ kernel in Au₂₃ is completely centrosymmetric (Supplementary Fig. 7a). The analysis of Au–Au distances between the Au₄ and Au₅ layers further confirms this conclusion (Supplementary Fig. 7b). The examination of the Au₄-Au₅-Au₄ sandwich kernel of **LC₁** indicates that its Au₁₃ kernel is slightly twisted (Supplementary Fig. 7c, d). Despite the tiny distortion of the Au₁₃ kernel in **LC₁**, the main difference between **LC₁** and Au₂₃ comes from the introduced phosphoramidites (**L₁**) ligands and the arrangements of -S-Au-S- and -S-Au-S-Au-S-Au-S- motifs on the surface of the Au₁₃ kernel. Specifically, Au₂₃ has four -S-Au-S- and two -S-Au-S-Au-S-Au-S- staple motifs (Fig. 3e–g), in which two of the -S-Au-S- motifs take the same arrangement as that in **LC₁**, bridging the upper and lower Au₄ layers of the Au₄-Au₅-Au₄ sandwich kernel (Fig. 3c, f). The other two -S-Au-S- and all of the -S-Au-S-Au-S-Au-S- motifs of Au₂₃ locate between the layers of the middle Au₅ and the upper (or lower) Au₄ (Fig. 3e, g). The total six staple motifs on the surface of the Au₁₃ kernel are also completely centrosymmetric in Au₂₃ (Fig. 3g). In contrast, the arrangement of -S-Au-S-Au-S-Au-S- motifs in **LC₁** is more diversified. One of the -S-Au-S-Au-S-Au-S- motifs twists around the middle Au₅ layer, and the other two locate between the middle Au₅ layer and the (upper or lower) Au₄ layer (Fig. 3d).

## Asymmetric transformation mechanism

Based on the experimental results and crystal structures of **LC₁** and Au₂₃, a proposed transformation mechanism from Au₂₃ to **LC₁** was shown in Supplementary Fig. 8. The two vertex Au atoms of the sandwich Au₁₃ kernel of Au₂₃ constitute two open sites (Supplementary Fig. 8a), which are easily attacked by the phosphine site of ligand **L₁**.

The Au–P interaction would trigger the cleavage of Au–S bonds and the further dissociation of the two corresponding -S-Au-S- staple motifs (Supplementary Fig. 8b). A subsequential rearrangement of a -S-Au-S-Au-S-Au-S- motif (Supplementary Fig. 8c) and the association of the two vacant Au sites with another -S-Au-S-Au-S-Au-S- would generate asymmetrically arranged staple motifs on the Au₁₃ kernel and give **LC₁** (Supplementary Fig. 8d). During the transformation, the Au₁₃ kernel is retained and only slightly twisted, while two short -S-Au-S- staple motifs are replaced by two phosphine ligands **L₁** and a long -S-Au-S-Au-S-Au-S- staple motifs. Therefore, **LC₁** contains one more Au atom and two **L₁** while possessing the same number of thiol ligands, when compared to Au₂₃. We carefully analyzed the product components of the reaction between Au₂₃ and **L₁** (Supplementary Table 2). Apart from a few undesired complexes and nanoparticles, **LC₁** was isolated as the major product with a yield of 65% (based on the Au atom). According to the whole process proposed in Supplementary Fig. 8, the mole ratio of the Au atom of two decomposed -S-Au-S- and one generated -S-Au-S-Au-S-Au-S- is 2/3, which is basically consistent with the yield (65%) of **LC₁** observed. This result suggests that 3 equivalents of Au₂₃ would produce 2 equivalents of **LC₁**, and the additional Au atom of **LC₁** probably comes from the decomposed -S-Au-S- staple motifs of Au₂₃. The Au₂₃ that are not transformed into **LC₁** during the reaction would decompose or aggregate into complexes or nanoparticles.

## Hierarchical chirality

Further analysis of the two enantiomers' structures reveals the origin of the chirality of the **LC₁** nanocluster (Fig. 4). First, (*R*)-**LC₁** and (*S*)-**LC₁** bear the same Au₁₃ kernel that can overlap completely (Supplementary Fig. 9), demonstrating that the Au₁₃ kernel is achiral. Second, the two short -S-Au-S- and three long -S-Au-S-Au-S-Au-S- motifs that surround

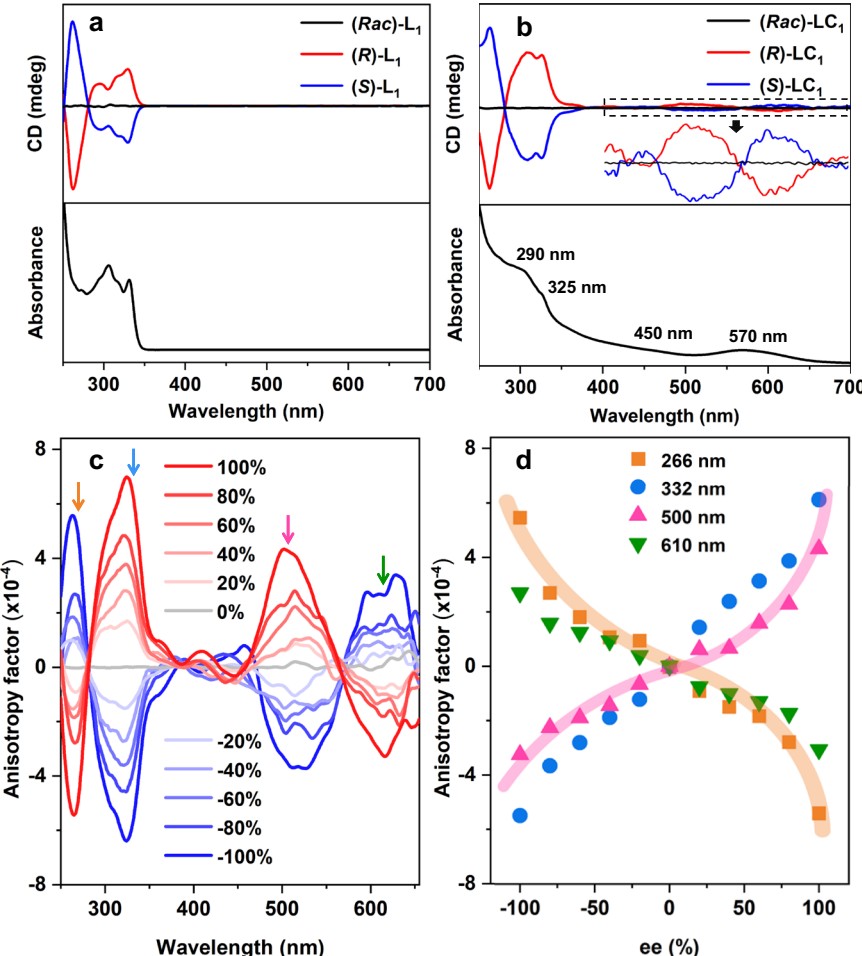

**Fig. 5 | Negative nonlinear dependence between chiroptical activity and enantiomeric excess. a** Combined CD-absorption spectra of (*Rac*)-$L_1$ (black trace), (*R*)-$L_1$ (red trace) and (*S*)-$L_1$ (blue trace). **b** Combined CD-absorption spectra of (*Rac*)-$LC_1$ (black trace), (*R*)-$LC_1$ (red trace) and (*S*)-$LC_1$ (blue trace). Inset: the enlarged view from 400 to 700 nm, and the magnification factor is 5. **c** Anisotropy factors of $LC_1$ prepared by different ee values of $L_1$. **d** The negative nonlinear CD-ee dependence between the chiroptical activity of $LC_1$ and enantiomeric excess of $L_1$ based on anisotropy. The solid and broader highlighted colored lines in (**d**) are merely guides to the eye. Source data are provided as a Source Data file.

around the sandwich $Au_{13}$ kernel of (*R*)-$LC_1$ cannot overlap with that of (*S*)-$LC_1$. Moreover, the arrangements of these motifs in (*R*)-$LC_1$ and (*S*)-$LC_1$ are completely mirror-symmetric (Fig. 4a, d), indicating that the chirality of $LC_1$ partially comes from the enantiotropic arrangement (tentatively defined as (*R*)- and (*S*)-arrangements) of surface motifs. Third, the (*R*)-$L_1$ and (*S*)-$L_1$ at the vertices of the $Au_{13}$ kernels of (*R*)-$LC_1$ and (*S*)-$LC_1$ are enantiomeric (Fig. 4b, e), which also contribute to the chirality of the nanocluster. More interestingly, we found that *R*)-$LC_1$ and (*S*)-$LC_1$ nanocluster molecules take the left-handed helix and right-handed helix arrangements, respectively, based on the observation of the crystallographic arrangements of the two enantiomeric nanoclusters (Supplementary Fig. 10). One of the helical and enantiotropic chains of (*R*)-$LC_1$ and (*S*)-$LC_1$ molecular arrangements were picked out and shown in Fig. 4c, f. The intermolecular hydrogen bonds might play an important role in the helical arrangement of nanocluster molecules (Supplementary Fig. 11). Based on the above analysis, $LC_1$ demonstrates a hierarchically chiral structure, which originates from three aspects: (1) the asymmetric arrangement of the -S-Au-S- and -S-Au-S-Au-S-Au-S- motifs on the surface of the $Au_{13}$ kernel; (2) the intrinsic chirality of phosphoramidite ligand at the two vertices of the $Au_{13}$ kernel; (3) the helical arrangement of nanocluster molecules. Thus, the phosphoramidite-induced asymmetric transformation from $Au_{23}$ to $LC_1$ not only introduces the intrinsic chirality of phosphoramidite itself

but also triggers the asymmetric rearrangements of the staple motifs and nanocluster molecules that were originally symmetric in $Au_{23}$.

## Negative nonlinear CD-ee dependence

The asymmetric transformation process is enantiodivergent, and the as-obtained (*R*)-$LC_1$ and (*S*)-$LC_1$ are enantiopure, displaying nearly perfect mirror-symmetric circular dichroism (CD) signals from 250 to 700 nm. Specifically, (*R*)-$LC_1$ gave positive Cotton effects at 310, 328, 405 and 500 nm with negative Cotton effects at 260, 425 and 600 nm, while (*S*)-$LC_1$ demonstrated the completely opposite Cotton effects (Fig. 5b). Based on the combined CD-absorption spectra, the zero-crossing points at 450 and 570 nm in the CD spectrum are consistent with the characteristic absorptions in the UV-vis spectrum, indicating that the exciton coupling happens. Compared with the chiral inducer (*R*)-$L_1$ and (*S*)-$L_1$ that only showed simple CD signals from 250 to 350 nm (Fig. 5a), the CD spectra of (*R*)-$LC_1$ and (*S*)-$LC_1$ displayed much more abundant signals. The anisotropy factors ($g = \Delta Abs/Abs$) of (*R*)-$L_1$ and (*S*)-$L_1$ were calculated in the wavelength ranging from 250 to 700 nm (Fig. 5c), showing a maximum value ($g_{max}$) of $0.75 \times 10^{-3}$ at 328 nm.

To gain a deeper insight into the asymmetric transformation of $Au_{23}$ induced by phosphoramidite, the relationship between the chiroptical activity of nanocluster $LC_1$ and the enantiomeric excess (ee) of

inducer $L_1$ was investigated. As shown in Fig. 5c, with the increment of ee values of $L_1$, the anisotropy of $LC_1$ increase accordingly. However, the CD-ee dependence is not linear based on the data collected at 266, 332, 500, and 610 nm of the anisotropy factors. Interestingly, we tried to obtain clusters containing one ($R$)-$L_1$ and one ($S$)-$L_1$ in the ligand shell by the reaction of $Au_{23}$ with ($Rac$)-$L_1$ (ee = 0%). However, the crystals suitable for single-crystal X-ray diffraction test were determined to be either ($R$)-$LC_1$ or ($S$)-$LC_1$. Considering that ($Rac$)-$LC_1$ showed none of the CD signal (Fig. 5b), the racemic clusters are likely composed of equimolar ($R$)-$LC_1$ and ($S$)-$LC_1$. This result is consistent with the nonlinear CD-ee dependence and suggests that the clusters with one ($R$)-$L_1$ and one ($S$)-$L_1$ in the ligand shell are thermodynamically unfavorable products. Notably, because other chiral nanoclusters than $LC_1$ exist in the crude product (Supplementary Table 2), the $LC_1$ nanocluster showed slightly different CD spectra from that of the crude product which was directly obtained from the reaction of $Au_{23}$ and $L_1$ (Supplementary Fig. 14). This might make the should-be linear CD-ee dependence change into nonlinear one. As we know, there are three types of CD-ee dependence for the chiral auxiliaries-induced chirality of materials: linear, positive nonlinear, and negative nonlinear[57]. The linear CD-ee dependence is quite common, and the positive nonlinear CD-ee dependence (also known as "majority rules effect") representing the chiral amplification phenomenon has only been found in a minority of the cases. In sharp contrast, the negative nonlinear CD-ee dependence has been much less reported. Herein, this dependence was found in metal nanoclusters. The CD-ee dependence of the asymmetric transformation process is nonlinear, indicating that more than one chiral auxiliary is involved in the intermediate of the asymmetric transformation of $Au_{23}$, providing support for the proposed mechanism (Supplementary Fig. 8). The negative nonlinear CD-ee dependence secures a steeper slope at the ee region close to 100%, which would be applied in the future for the accurate determination of enantiopurity of molecules at the high ee region[58]. Such determination is important in practical applications, such as the optimization of asymmetric catalyst performance.

### Asymmetric transformation application

Apart from the introduction of hierarchical chirality, the phosphoramidite-induced transformation of $Au_{23}$ also leads to the improvement of stability and photoluminescence of the gold nanocluster. Time-dependent UV-vis spectra showed that $LC_1$ was stable even at 80 °C under the ambient atmosphere (Supplementary Fig. 15b). In contrast, $Au_{23}$ was quickly decomposed under the same conditions (Supplementary Fig. 15a). Based on the molecular formula ($Au_{24}(L_1)_2(SC_6H_{11})_{16}$) of $LC_1$, it bears an eight-electron closed-shell $(24 \times 1 - 16 \times 1 = 8)$ structure. Moreover, differential pulse voltammetry (dpv) reveals that the oxidation (0.22 V) and reduction (−1.65 V) barriers of $LC_1$ are higher than that of $Au_{23}$ (0.07 V and −1.34 V, Supplementary Fig. 16). The stable electronic structure and the relatively large electrochemical gap all contribute to the high stability of $LC_1$. The fluorescence emission peaks of $Au_{23}$ and $LC_1$ locate at a similar wavelength of 720 nm based on the photoluminescence spectra (Supplementary Fig. 17). However, the emission intensity of $LC_1$ is almost five times higher than that of $Au_{23}$. $LC_1$ dissolved in DCM displayed stronger red emission than $Au_{23}$ by keeping their concentrations the same (Supplementary Fig. 17, inset). Considering that $LC_1$ and $Au_{23}$ bear the basically same $Au_{13}$ kernel, the fluorescence enhancement probably originates from the asymmetric transformation-resulted structural modification on the surface.

As mentioned above, phosphoramidite is modular and easy to synthesize. Using different amino modules involving sterically hindered, unsymmetric, aryl and alkyl groups, we synthesized the other four enantiopure phosphoramidites ($L_{2-5}$), which were applied to induce the asymmetric transformations of $Au_{23}$. These phosphoramidites all reacted well with $Au_{23}$, affording $LC_{2-5}$ with racemic, R and

S configurations (Fig. 6). The exciton-coupled CD profiles with respect to the UV-vis spectra of $LC_1$-$LC_5$ were carefully analyzed. Based on the combined CD-absorption spectra (Supplementary Fig. 12a–e), $LC_1$, $LC_2$, $LC_3$ and $LC_5$ demonstrate characteristic absorptions at 290, 325 and 570 nm, and a weak peak at about 450 nm. Accordingly, these four clusters showed consistent zero-crossing points at 450 and 570 nm in the CD profiles. $LC_4$ showed a slightly different CD spectrum. This can be explained by a different UV-vis spectrum of ligand $L_4$, compared to that of the other four phosphoramidite ligands (Supplementary Fig. 12f). The distinctive absorption of $L_4$ might originate from the conjugated π system of the two phenyl groups on the nitrogen atom. The UV-vis spectrum of $LC_4$ demonstrates a stronger absorption peak at 410 nm and a bathochromic shift at 580 nm, which are also slightly different from that of $LC_1$, $LC_2$, $LC_3$ and $LC_5$. The absorption at 410 nm is consistent with an obvious Cotton effect at this wavelength. ESI-MS spectra of these nanoclusters confirmed their molecular formulas to be $Au_{24}(L_{2-5})_2(SC_6H_{11})_{16}$ (Fig. 6a–d). The above analysis combined with the ESI-MS spectra suggest that $LC_1$-$LC_5$ have similar structures.

## Discussion

In summary, we realized the asymmetric transformation of an achiral metal nanocluster in this work for the achievement of enantiopure metal nanoclusters. Phosphoramidites ($L$) were developed as efficient chiral auxiliaries to induce the enantiodivergent processes, and the enantiomeric nanoclusters ($R$)- and ($S$)-$Au_{24}(L)_2(SC_6H_{11})_{16}$ were synthesized separately from achiral $Au_{23}$. Structural analysis reveals that the asymmetric transformation not only brings the intrinsic chirality of phosphoramidite but also triggers the asymmetric rearrangement of the staple motifs and nanocluster molecules that were originally symmetric in $Au_{23}$, constituting the hierarchical chirality of $Au_{24}(L)_2(SC_6H_{11})_{16}$. A negative nonlinear CD-ee dependence was found for the relationship between the chiroptical activity of $Au_{24}(L)_2(SC_6H_{11})_{16}$ and the enantiomeric excess of $L$. The phosphoramidite is modular and a series of substituents can be introduced, leading to the functional diversity of the as-transformed chiral nanoclusters. We expect that our work will stimulate further research on the construction of enantiopure metal nanoclusters and the chirality of complex systems.

## Methods

### Characterizations

ESI-MS were acquired on a Waters Q-TOF mass spectrometer equipped with a Z-spray source. All UV-vis absorption measurements were performed on a SPECORD 210 PLUS spectrophotometer. SCXRD data were measured by using a Stoe Stadivari diffractometer. The structures were solved and refined using the SHELXT software. CD spectra were obtained by Circular chromatograph J-1700, and the reference solvent for measurement is DCM. Fluorescence spectra were obtained by a spectrofluorometer FS 5. Electrochemical measurements were performed with a CHI770E electrochemistry workstation in a three-electrode system using an Ag/AgCl electrode as the reference electrode, a glassy carbon electrode as the working electrode and a platinum wire electrode as the auxiliary electrode. NMR spectra were recorded at 400 MHz. Chemical shifts (δ) are reported in ppm, using the residual solvent peak in CDCl$_3$ (7.26 ppm) as the internal standard. Coupling constants ($J$) are given in Hz.

### Synthesis of $Au_{23}$

HAuCl$_4$·3H$_2$O (0.3 mmol, 118 mg) and tetraoctylammonium bromide (TOAB, 0.35 mmol, 190 mg) were dissolved in methanol (15 mL) in a 100 mL round-bottom flask. After vigorously stirring for 15 min, cyclohexanethiol (1.6 mmol, 196 μL) was added to the mixture at room temperature. After 15 min, NaBH$_4$ (3 mmol, 114 mg dissolved in 6 mL of cold Nanopure water) was rapidly added to the solution under

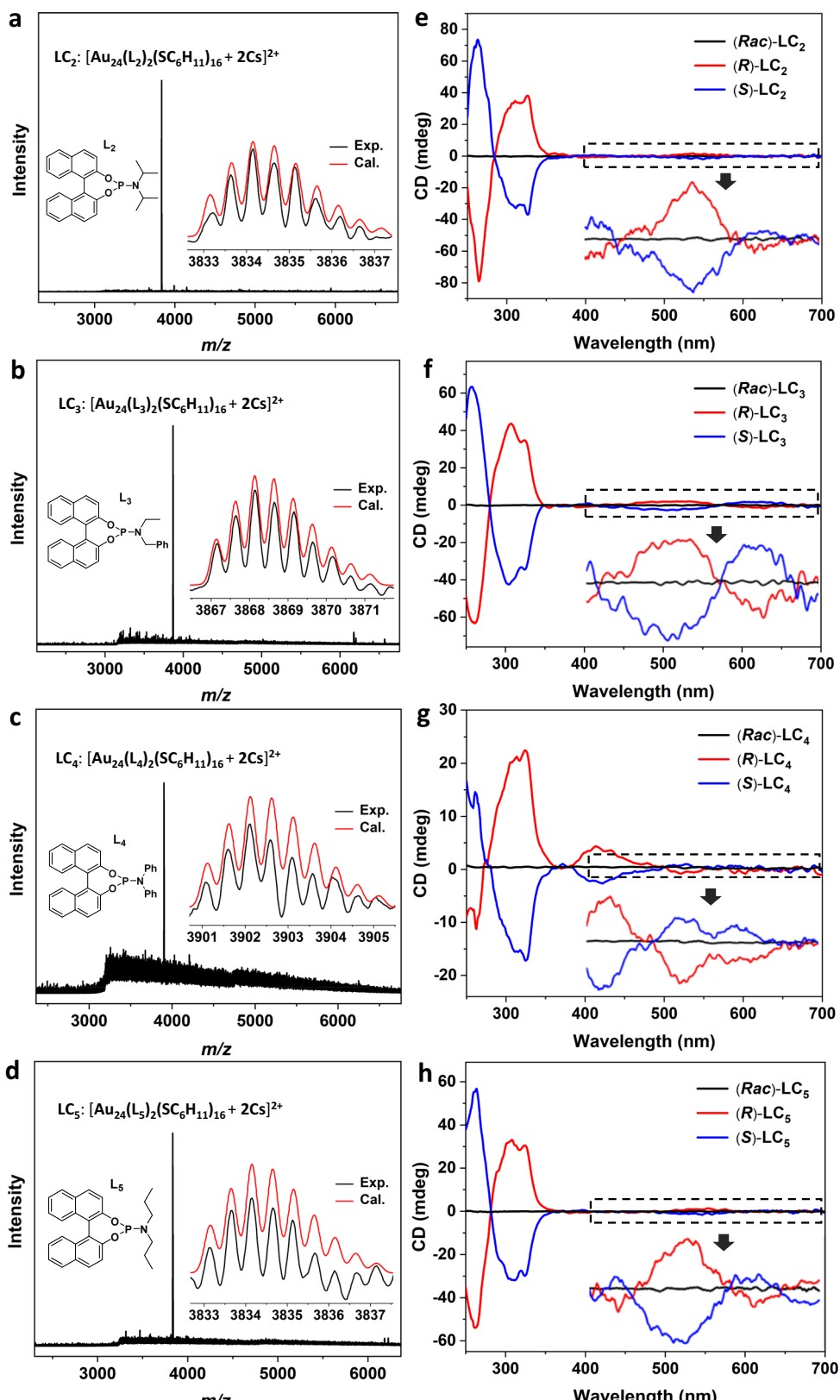

**Fig. 6 | Characterizations of LC$_{2-5}$. a–d** ESI-MS spectra of **LC$_{2-5}$**. Inset: the structures of **L$_{2-5}$**, and the experimental (black trace) and calculated (red trace) isotope patterns of **LC$_{2-5}$**. **e–h** CD spectra of (**Rac**)- (black trace), (**R**)- (red trace), and (**S**)-**LC$_{2-5}$** (blue trace). Inset: the enlarged view of CD spectra from 400 to 700 nm, the magnification factors of **LC$_{2-5}$** are 5, 4, 3 and 6. Source data are provided as a Source Data file.

vigorous stirring. The reaction mixture was allowed to stir overnight and finally gave Au$_{23}$ in 20% yield (purified by recrystallization).

## Preparation of phosphoramidite (L$_{1-5}$)

By using the procedure for the preparation of (S)-L$_1$ as an example: Triethylamine (5.0 eq., 25 mmol, 3.5 mL) was added dropwise to a stirred ice-cooled solution of PCl$_3$ (5 mmol, 436 μL) in CH$_2$Cl$_2$ (35 mL). The ice bath was removed and the solution was warm to room temperature before diethylamine (5 mmol, 517 μL) was added. After 5 h, (S)-binaphthol (5 mmol, 1.43 g) was added to the suspension and the resulting mixture was left to stir for an additional 18 h. The purification of (S)-L$_1$ was achieved via column chromatography on silica gel (eluent: Pentane/EtOAc = 60/1). (Rac)-, (R)- and (S)-L$_{1-5}$ were prepared following the procedure above.

## Synthesis of LC$_{1-5}$ from Au$_{23}$

By using the procedure for the synthesis of (S)-LC$_1$ as an example: 5.0 equivalents of (S)-L$_1$ (4.24 mg) were added into the CH$_2$Cl$_2$ (1.5 mL) solution of Au$_{23}$ (15.0 mg). After the completion of the reaction in 4 h, the mixture was washed with MeOH for 2-3 times to give the crude product (S)-LC$_1$ (precipitate), which was recrystallized in the system of CH$_2$Cl$_2$/n-hexane. Red hexagon crystals were obtained after 7 days, which were suitable for ESI-MS, SCXRD and the other characterizations. (Rac)-, (R)- and (S)-LC$_{1-5}$ were prepared following the procedure above.

## CD-ee dependence studies

First, L$_1$ with different ee (−100%, −80%, −60%, −40%, −20%, 0%, 20%, 40%, 60%, 80% and 100%) were prepared by mixing optical pure (R)-L$_1$ and (S)-L$_1$. The amount of (R)-L$_1$ and (S)-L$_1$ used for each sample was calculated via Eq. (1) and is also shown in Supplementary Table 3. Second, 5.0 equivalents of L$_1$ (5.0 mg) with specific ee was added into 18.0 mg of Au$_{23}$ (dissolved in 1.5 mL of DCM). After 4 h, the reaction was purified by column chromatography on silica gel (eluent: DCM/MeOH = 20/1) to give LC$_1$. Third, the purified LC$_1$ from the reaction of Au$_{23}$ and L$_1$ with different ee was used for the CD test. The corresponding anisotropy factors (g) were obtained via Eq. (2). θ and Abs refer to ellipticity and absorbance, respectively.

$$ee = \frac{[R] - [S]}{[R] + [S]} \times 100\% \tag{1}$$

$$g = \frac{\theta / mdeg}{32{,}980 \times Abs} \tag{2}$$

## Stability studies

For this, 5.0 mg of pure Au$_{23}$ or LC$_1$ was dissolved in 2 mL of toluene. The solution was gently stirred at 80 °C. The time-dependent UV-vis absorption spectra were obtained based on the mixture (Supplementary Fig. 15). 5.0 mg of Au$_{23}$ or LC$_1$ was dissolved in Bu$_4$NPF$_6$ (70 mg)-DCM (2 mL) solution and the electrochemical property of the nanocluster was measured using an electrochemical workstation. Before the experiment, the working electrode was polished with a mixture of Al$_2$O$_3$ and water and then cleaned sequentially with water and MeOH. The experiment was performed at an amplitude of 0.05 V, a pulse width of 0.05 s, a sampling width of 0.02 s and a pulse period of 0.1 s. The sample was always in a nitrogen atmosphere.

## Photoluminescence studies

The excitation and emission spectra of Au$_{23}$ and LC$_1$ were obtained by dissolving the nanocluster in DCM at room temperature. The concentration of the samples was kept at the same to be 2.3 × 10$^{-5}$ M. The excitation wavelength was kept at 350 and 360 nm, respectively, for the emission spectra of Au$_{23}$ and LC$_1$. The data were collected based on the same parameters.

## Reporting summary

Further information on research design is available in the Nature Portfolio Reporting Summary linked to this article.

## Data availability

The data that support the findings of this study are available from the corresponding author upon request. Source data are provided with this paper. The X-ray crystallographic structures reported in this work have been deposited at the Cambridge Crystallographic Data Center (CCDC) under deposition numbers 2216153 and 2216156 for (R)- and (S)-Au$_{24}$(L$_1$)$_2$(SC$_6$H$_{11}$)$_{16}$, respectively. These data can be obtained free of charge from the CCDC via https://www.ccdc.cam.ac.uk/structures/. Source data are provided with this paper.

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

## Acknowledgements

M.-B.L. acknowledges the financial support from the National Natural Science Foundation of China (92061110), the Anhui Provincial Natural Science Foundation (2108085Y05), the Innovation and Entrepreneurship Project of Overseas Returnees in Anhui Province (2022LCX014), and the Hefei National Laboratory for Physical Sciences at the Microscale (KF2020102).

## Author contributions

M.-B.L. supervised the research and wrote the manuscript. M.-B.L. and S.Z. summarized the data. C.L., T.-S.Z. and C.-B.T. carried out the experiments. Y.Z. and W.F. resolved the structures of metal nanoclusters. All authors contributed to the preparation of the manuscript.

## Competing interests

The authors declare no competing interests.
