## [Peer Review File · Nature Communications]

Reviewer comments, first round

Reviewer #1 (Remarks to the Author):

This is an interesting study that reveals the transformation of an achiral cluster towards a chiral one upon addition of a chiral ligand. The authors could determine the structure of the involved clusters and then also reveal a “negative nonlinear CD ee dependence”. Also, the authors observe a helical arrangement of the clusters in the crystal. Overall these results are of great value to the field as chiral enantiopure clusters have good potential for applications.

The authors should clarify some points before publication:

The authors write in the introduction: “... two methods were applied for acquiring metal nanoclusters with enantiomeric excess: (i) enantioseparation of chiral nanoclusters (racemic) based on chiral high performance liquid chromatography (HPLC) or resolving agents; (ii) direct synthesis of enantiopure nanoclusters by using chiral organic ligands...” There is at least another promising method: enantioselective phase transfer, see the work of Knoppe (JACS, 136 (11), 4129-4132, 2014) Right after that the authors write: “...The former method is limited to chiral nanoclusters, and ...” This sentence does not make sense because all methods for the separation of enantiomers require chiral samples...else there are no enantiomers. Please clarify this.

The newly formed cluster has one more metal atom. Where does this come from? Likely, during the transformation some of the original cluster is decomposed. The authors should comment on that.

Also, can the authors estimate a yield of the new cluster?

Do the authors obtain clusters with one R and one S ligand in the ligand shell?

For the experiments on “negative nonlinear CD ee dependence” the authors use the CD signal, which however concentration dependent. I propose to use anisotropy instead of ellipticity (“CD signal”) in these plots.

Reviewer #2 (Remarks to the Author):

See attached file.

Reviewer #3 (Remarks to the Author):

This work reports the use of R- and S-phosphoramidites as chiral ligands to induce the transformation of achiral Au₂₃ into enantiomeric nanoclusters of Au₂₄(L)₂(SC₆H₁₁)₁₆ (where L = chiral phosphoramidite). X-ray diffraction, CD spec and other characterizations were performed.

After reading, I do not feel that the results are exciting enough for publication in Nat Commun.

The chiral ligand-induced synthesis is quite well known, and the Au₂₄ enantiomers exhibit weak CD ($g < 10^{-3}$). The origin of chirality (e.g. arrangements of staple motifs) is also well known.

Overall, this work would not meet the expectations of Nat Commun.

This is an interesting piece of work describing a method for synthesizing chiral Au-nanoclusters from an achiral precursor which is allowed to react with a chiral ligand. This forms the most important feature of this work that the chiral species is obtained by reacting an achiral precursor with a chiral ligand. This may in the future be developed into a general protocol if other achiral Au-nanoclusters are also applicable for the reaction. The present work has shown that it is applicable for the same achiral Au-nanoclusters with a variety of the chiral ligands differing slightly in the structural moiety that does not react with the metal nanoclusters. The obtained chiral nanoclusters are well characterized by X-ray crystallography and spectral technique, absorption, CD and emission. While the results and the synthetic strategy are of significance, several issues shall be well addressed before it is accepted for publication in Nature Communications.

1. The synthesized chiral Au-nanoclusters contains one more Au atom(ion) while with the same number of thiol ligand, when compared to those in the achiral metal nanocluster precursor. This is an issue not discussed at all. A detailed discussion on this issue however would help clarify the mechanism for the "reaction". I thus suggest authors to discuss in detail on this issue. For this purpose, it is suggested to report the reaction yields, to help identify the "origin" of this additional Au-element. This requirement also applies to the other chiral nanoclusters obtained using other chiral ligands (L2-L5).

2. As for the synthesis of chiral nanoclusters the statement on page 2 lines -2 to -1 ", and has not been previously studied..." is not fully true, since for example pure chiral metal nanoclusters were obtained using a reduction reaction with metal species, work reported in reference 28 for example.

3. The mechanism has been discussed on the basis of the crystal structures, which shows that the two bulky chiral ligand molecules only respectively attach to one Au-species, the arrangement of the staple structures which are in close proximity are made chiral while the central kernel structure is only slightly twisted. The general structure of the metal nanocluster precursor is suggested to be minorly modified by the binding of the chiral ligand, that the absorption spectra of the chiral metal nanoclusters are more or less the same as that of the achiral precursor Au₂₃ (Fig 1b). A detailed analysis of the CD profiles with respect to the absorption spectra (Figures 1b and 5a-c) would help. Look at CD spectra shown in Figure 5a and 5b, we note that the major CD signals originate from the chiral ligand, below 350 nm, while those from the metal nanoclusters, showing absorption around 570 nm (Figure 1b), between 400-700 nm are very weak. The CD signals of the chiral Au-nanoclusters obtained from chiral ligand L1 is exciton-coupled, from a comparison of CD spectra enlarged part in Fig 5b showing a zero-crossing point at 560 nm, the same as the absorption maximum shown in Fig 1b. This conclusion is not all true for those chiral nanoclusters from other ligands (Fig 6). It appears only likely with that from L3 with a structure similar to that of L1. The discussion of this exciton-coupled CD signal however could help much for the understanding of the chiral structure when the chromophores responsible for the absorption at 560 nm are well identified. The CD-ee dependence obtained using CD signals at ligand absorption region may only reflect the chiral preference of the ligand molecules in the nanoclusters. It is thus suggested to use CD signals at 500 and 600 nm to build the CD-ee curves. I would also ask the authors to describe in the "Experimental" part how the chiral nanoclusters are prepared for building the CD-ee curves, using enantiomeric mixtures of the ligand of varying ee to

react with the achiral metal nanocluster precursor, or mixing the chiral R-/S-nanoclusters respectively prepared.

Reviewer 1:

This is an interesting study that reveals the transformation of an achiral cluster towards a chiral one upon addition of a chiral ligand. The authors could determine the structure of the involved clusters and then also reveal a “negative nonlinear CD ee dependence”. Also, the authors observe a helical arrangement of the clusters in the crystal. Overall these results are of great value to the field as chiral enantiopure clusters have good potential for applications. The authors should clarify some points before publication.

Response: We thank the reviewer for the positive comments.

1. The authors write in the introduction: “... two methods were applied for acquiring metal nanoclusters with enantiomeric excess: (i) enantioseparation of chiral nanoclusters (racemic) based on chiral high performance liquid chromatography (HPLC) or resolving agents; (ii) direct synthesis of enantiopure nanoclusters by using chiral organic ligands...” There is at least another promising method: enantioselective phase transfer, see the work of Knoppe (JACS, 136 (11), 4129-4132, 2014). Right after that the authors write: “...The former method is limited to chiral nanoclusters, and” This sentence does not make sense because all methods for the separation of enantiomers require chiral samples...else there are no enantiomers. Please clarify this.

Response: We thank the reviewer for pointing out the inaccurate description and advising the quite important and useful paper. We have given a more comprehensive discussion with reference to alternative approaches and previous reports in the text (p. 2, highlight lines) of the revised manuscript, which is also shown below:

“Based on the previous reports, some elegant methods were developed toward acquiring metal nanoclusters with enantiomeric excess. These methods include: (i) enantioseparation of racemic mixtures of metal nanoclusters by chiral high performance liquid chromatography (HPLC)^[21-24] or resolving agents;^[25-27] (ii) enantioselective phase transfer of chiral nanoclusters;^[28,29] (iii) direct synthesis of enantiopure nanoclusters by using chiral organic ligands as the precursors.^[30-37] In spite of the significant progress, the method for transforming an achiral metal nanocluster into a chiral one is limited. There are a few reports on the chiral ligand-exchange of achiral metal nanoclusters,^[38,39] in which the chirality generally comes from the chiral ligands on the surface. In view of hundreds of achiral metal nanoclusters being reported^[13] and the wide application of asymmetric organic transformations in pharmaceutical synthesis and industry chemistry,^[40,41] we envisioned the possibility of using a chiral inducer to realize the asymmetric transformation of achiral metal nanoclusters. This strategy would open up a new opportunity and be developed into a general protocol for the construction of enantiopure metal nanoclusters.”

The related papers were added in ref. 22, 25, 28, 29, 38 and 39 of the revised manuscript.

2. The newly formed cluster has one more metal atom. Where does this come from? Likely, during the transformation some of the original cluster is decomposed. The authors should comment on that. Also, can the authors estimate a yield of the new cluster?

Response: We thank the reviewer for the advice. We have conducted careful experiments and isolated the newly formed cluster by column chromatography on silica gel (eluent: DCM/MeOH = 20/1). The isolated yield of the newly formed cluster $\text{Au}_{24}(\text{L}_1)_2(\text{SC}_6\text{H}_{11})_{16}$ (LC_1) was determined to be 65% (based on Au atom). Based on the experimental results and crystal structures of LC_1 and Au_{23} , we commented on the transformation process and the origin of the additional Au atom in the text (p. 7 and 8, highlight lines) of the revised manuscript.

We propose that some of the staple motifs on the surface of Au_{23} are decomposed, while the Au_{13} kernel is basically unaffected during the transformation. Specifically, the interaction between ligand L_1 and Au_{23} would trigger the cleavage of Au–S bonds on the two vertex Au atoms of the Au_{13} kernel and the further decomposition of the two corresponding -S-Au-S- staple motifs. A subsequential rearrangement of a -S-Au-S-Au-S-Au-S- motif and the association of another -S-Au-S-Au-S-Au-S- would finally give LC_1 . From the whole process, two short -S-Au-S- staple motifs are decomposed while a long -S-Au-S-Au-S-Au-S- staple motifs is generated. Therefore, LC_1 contains one more Au atom than Au_{23} . The mole ratio of Au atom of two decomposed -S-Au-S- and one generated -S-Au-S-Au-S-Au-S- is 2/3, which is basically consistent with the yield (65%) of LC_1 observed. This result suggests that 3 equivalents of Au_{23} would produce 2 equivalents of LC_1 , and the one more Au atom of LC_1 probably comes from the decomposed -S-Au-S- staple motifs of Au_{23} . The Au_{23} that are not transformed into LC_1 during the reaction would decompose or aggregate into complexes or nanoparticles, which were also observed and isolated.

The corresponding comment was added in the revised manuscript. The yield of the new cluster and the proposed transformation process were added in Table S2 (p. S6) and Figure S8 (p. S11) of the revised Supporting Information.

3. Do the authors obtain clusters with one R and one S ligand in the ligand shell?

Response: We thank the reviewer for the question. By using (*R*)- L_1 or (*S*)- L_1 as the chiral inducer, only (*R*)- LC_1 or (*S*)- LC_1 was obtained, in which either two (*R*)- L_1 or two (*S*)- L_1 are in the ligand shell. We have prepared the racemic cluster (*Rac*)- LC_1 by the reaction of Au_{23} with (*Rac*)- L_1 , and tried to obtain clusters containing one R and one S ligand in the ligand shell. However, the crystals suitable for single-crystal X-ray diffraction test were determined to be either (*R*)- LC_1 or (*S*)- LC_1 . Considering that (*Rac*)- LC_1 showed none of CD signal (Figure 5b), the racemic clusters are likely composed of equimolar (*R*)- LC_1 and (*S*)- LC_1 . This result is consistent with the nonlinear CD-ee dependence, and suggests that the clusters with one (*R*)- L_1 and one (*S*)- L_1 in the ligand shell are thermodynamically unfavorable products. The corresponding comments have been added in the text (p. 10 and 11, highlight lines) of the revised manuscript.

4. For the experiments on “negative nonlinear CD ee dependence” the authors use the CD signal, which however concentration dependent. I propose to use anisotropy instead of ellipticity (“CD signal”) in these plots.

Response: We thank the reviewer for the advice. We have rebuilt the CD-ee dependence based on anisotropy instead of ellipticity. The corresponding curves were presented in Figure 5c of the revised manuscript, and also shown below:

Reviewer 2:

This is an interesting piece of work describing a method for synthesizing chiral Au-nanoclusters from an achiral precursor which is allowed to react with a chiral ligand. This forms the most important feature of this work that the chiral species is obtained by reacting an achiral precursor with a chiral ligand. This may in the future be developed into a general protocol if other achiral Au-nanoclusters are also applicable for the reaction. The present work has shown that it is applicable for the same achiral Au-nanoclusters with a variety of the chiral ligands differing slightly in the structural moiety that does not react with the metal nanoclusters. The obtained chiral nanoclusters are well characterized by X-ray crystallography and spectral technique, absorption, CD and emission. While the results and the synthetic strategy are of significance, several issues shall be well addressed before it is accepted for publication in Nature Communications.

Response: We thank the reviewer for the positive comments.

1. The synthesized chiral Au-nanoclusters contains one more Au atom(ion) while with the same number of thiol ligand, when compared to those in the achiral metal nanocluster

precursor. This is an issue not discussed at all. A detailed discussion on this issue however would help clarify the mechanism for the "reaction". I thus suggest authors to discuss in detail on this issue. For this purpose, it is suggested to report the reaction yields, to help identify the "origin" of this additional Au-element. This requirement also applies to the other chiral nanoclusters obtained using other chiral ligands (L₂-L₅).

Response: We thank the reviewer for the advice. We have systematically analyzed the product components of the reaction between Au₂₃ and phosphoramidite (L₁-L₅). Component A (soluble complexes), component B (LC_{1,5}) and component C (insoluble nanoparticles) were observed and their yields were reported in Table S2 of the revised Supporting Information (p. S6). Based on these experimental results and crystal structures of LC₁ and Au₂₃, a detailed discussion on the mechanism of the reaction and the origin of the additional Au-element was added in the text (p. 7 and 8, highlight lines) of the revised manuscript and Figure S8 (p. S11) of the revised Supporting Information. The detailed discussion, Table S2, and Figure S8 are also shown below:

“Based on the experimental results and crystal structures of LC₁ and Au₂₃, a proposed transformation mechanism from Au₂₃ to LC₁ was shown in Figure S8. The two vertex Au atoms of the sandwich Au₁₃ kernel of Au₂₃ constitute two open sites (Figure S8a), which are easily attacked by the phosphine site of ligand L₁. The Au–P interaction would trigger the cleavage of Au–S bonds and the further dissociation of the two corresponding -S-Au-S- staple motifs (Figure S8b). A subsequential rearrangement of a -S-Au-S-Au-S-Au-S- motif (Figure S8c) and the association of the two vacant Au sites with another -S-Au-S-Au-S-Au-S- would generate asymmetrically arranged staple motifs on the Au₁₃ kernel and give LC₁ (Figure S8d). During the transformation, the Au₁₃ kernel is retained and only slightly twisted, while two short -S-Au-S- staple motifs are replaced by two phosphine ligands L₁ and a long -S-Au-S-Au-S-Au-S- staple motifs. Therefore, LC₁ contains one more Au atom and two L₁ while possesses the same number of thiol ligand, when compared to Au₂₃. We carefully analyzed the product components of the reaction between Au₂₃ and L₁ (Table S2). Apart from a few undesired complexes and nanoparticles, LC₁ was isolated as the major product with the yield of 65% (based on Au atom). According to the whole process proposed in Figure S8, the mole ratio of Au atom of two decomposed -S-Au-S- and one generated -S-Au-S-Au-S-Au-S- is 2/3, which is basically consistent with the yield (65%) of LC₁ observed. This result suggests that 3 equivalents of Au₂₃ would produce 2 equivalents of LC₁, and the additional Au atom of LC₁ probably comes from the decomposed -S-Au-S- staple motifs of Au₂₃. The Au₂₃ that are not transformed into LC₁ during the reaction would decompose or aggregate into complexes or nanoparticles.”

Table S2. Reaction results of L₁-L₅ with Au₂₃

	$\text{Au}_{23}(\text{SC}_6\text{H}_{11})_{16}$ 15 mg	$\xrightarrow[\text{DCM, rt, 4 h}]{\text{L}_{1-5} \text{ (5.0 equiv.)}}$	$\text{Au}_{24}(\text{L})_2(\text{SC}_6\text{H}_{11})_{16}$ LC₁₋₅
	Yields of products		
L₁₋₅	Component A (complexes)	Component B (LC₁₋₅)	Component C (nanoparticles)
L₁	3.0 mg	10.0 mg, 65%	1.3 mg
L₂	3.8 mg	9.20 mg, 59%	1.5 mg
L₃	3.3 mg	10.2 mg, 65%	1.3 mg
L₄	3.0 mg	9.00 mg, 57%	1.5 mg
L₅	3.5 mg	10.0 mg, 64%	1.4 mg

Figure S8. The proposed transformation mechanism from Au_{23} to LC_1 .

2. As for the synthesis of chiral nanoclusters the statement on page 2 lines -2 to -1 ", and has not been previously studied..." is not fully true, since for example pure chiral metal nanoclusters were obtained using a reduction reaction with metal species, work reported in reference 28 for example.

Response: We thank the reviewer for pointing out this. We have deleted the inaccurate statement and given a more comprehensive discussion with reference to alternative approaches and previous reports in the text (p. 2, highlight lines) of the revised manuscript.

3. The mechanism has been discussed on the basis of the crystal structures, which shows that the two bulky chiral ligand molecules only respectively attach to one Au-species, the arrangement of the staple structures which are in close proximity are made chiral while the central kernel structure is only slightly twisted. The general structure of the metal nanocluster precursor is suggested to be minorly modified by the binding of the chiral ligand, that the absorption spectra of the chiral metal nanoclusters are more or less the same as that of the achiral precursor Au_{23} (Fig 1b). A detailed analysis of the CD profiles with respect to the absorption spectra (Figures 1b and 5a-c) would help. Look at CD spectra shown in Figure 5a and 5b, we note that the major CD signals originate from the chiral ligand, below 350 nm, while those from the metal nanoclusters, showing absorption around 570 nm (Figure 1b),

between 400-700 nm are very weak. The CD signals of the chiral Au-nanoclusters obtained from chiral ligand L₁ is exciton-coupled, from a comparison of CD spectra enlarged part in Fig 5b showing a zero-crossing point at 560 nm, the same as the absorption maximum shown in Fig 1b. This conclusion is not all true for those chiral nanoclusters from other ligands (Fig 6). It appears only likely with that from L₃ with a structure similar to that of L₁. The discussion of this exciton-coupled CD signal however could help much for the understanding of the chiral structure when the chromophores responsible for the absorption at 560 nm are well identified. The CD-ee dependence obtained using CD signals at ligand absorption region may only reflect the chiral preference of the ligand molecules in the nanoclusters. It is thus suggested to use CD signals at 500 and 600 nm to build the CD-ee curves. I would also ask the authors to describe in the “Experimental” part how the chiral nanoclusters are prepared for building the CD-ee curves, using enantiomeric mixtures of the ligand of varying ee to react with the achiral metal nanocluster precursor, or mixing the chiral R-/S-nanoclusters respectively prepared.

Response:

(1) We thank the reviewer for the advice of detailed analysis of the exciton-coupled CD profiles with respect to the UV-vis spectra. For this purpose, we carefully reconducted the experiments and obtained clear CD and UV-vis spectra of LC₁-LC₅. The combined CD-absorption spectra were added in Figure S12 of the revised Supporting Information, and also shown below. The Figure 6 has also been modified accordingly. Based on the spectra (Figure S12a-e), LC₁, LC₂, LC₃ and LC₅ demonstrate characteristic absorptions at 290, 325 and 570 nm, and a weak peak at about 450 nm. Accordingly, these four clusters showed consistent zero-crossing points at ~ 450 and 570 nm in the CD profiles. LC₄ showed a slightly different CD spectrum. This can be explained by a different UV-vis spectrum of ligand L₄, compared to that of the other four phosphoramidite ligands (Figure S12f). The distinctive absorption of L₄ might originate from the conjugated π system of the two phenyl groups on the nitrogen atom. The UV-vis spectrum of LC₄ demonstrates a stronger absorption peak at ~ 410 nm and a bathochromic shift at 580 nm, which are also slightly different from that of LC₁, LC₂, LC₃ and LC₅. The absorption at ~ 410 nm is consistent with an obvious Cotton effect at this wavelength. The above analysis, combined with the ESI-MS spectra in Figure 6 suggest that LC₁-LC₅ have the similar structures. The corresponding comment has also been added in the text (p. 12, highlight lines) of the revised manuscript.

Figure S12. (a-e) Combined CD-absorption spectra of LC₁–LC₅. (f) UV-vis spectra of L₁–L₅

(2) We used enantiomeric mixtures of L₁ of varying ee to react with the achiral metal nanocluster Au₂₃ for building the CD-ee curves. The description has been added in the

“methods” part (p. 15, highlight lines) of the revised manuscript. The details of the experiments have also been added in the revised Supporting Information (p. S14-S15). We have used the signals at 500 and 610 nm to build the CD-ee dependence. Moreover, we have rebuilt the CD-ee dependence based on anisotropy instead of ellipticity to avoid the impact of concentration according to the suggestion of another reviewer. The results showed that the CD-ee dependence at 500 and 610 nm is consistent with that at 266 and 322 nm to be negative nonlinear. The corresponding curves have been shown in Figure 5c of the revised manuscript and Figure S13 of the revised Supporting Information. Figure S13 is also shown below:

Figure S13. (a) Anisotropy factors of LC_1 prepared by different ee values of L_1 . (b and c) Representation of the negative nonlinear CD-ee dependence at 266, 322, 500 and 610 nm.

Reviewer 3:

This work reports the use of R- and S-phosphoramidites as chiral ligands to induce the transformation of achiral Au_{23} into enantiomeric nanoclusters of $Au_{24}(L)_2(SC_6H_{11})_{16}$ (where L = chiral phosphoramidite). X-ray diffraction, CD spec and other characterizations were performed. After reading, I do not feel that the results are exciting enough for publication in

Nat Commun. The chiral ligand-induced synthesis is quite well known, and the Au₂₄ enantiomers exhibit weak CD ($g < 10^{-3}$). The origin of chirality (e.g. arrangements of staple motifs) is also well known. Overall, this work would not meet the expectations of Nat Commun.

Response: We thank the reviewer for the comment. Although the direct synthesis of metal nanoclusters by the use of chiral ligands has been developed (see the revised manuscript: p. 2, method iii in highlight lines), the report on asymmetrically structural transformation of an achiral metal nanocluster into a chiral one is limited. The most important feature of this work is that the enantiopure metal nanoclusters are obtained by the reaction of an achiral metal nanocluster with a chiral ligand. This may in the future be developed into a general protocol for the construction of enantiopure metal nanoclusters. The as-obtained nanoclusters show hierarchical chirality, which originates from three aspects: (i) the introduced chiral ligands at the two vertices of the Au₁₃ kernel; (ii) the transformation-triggered asymmetric rearrangement of the staple motifs; (iii) the helical arrangement of nanocluster molecules. The hierarchical chirality might enable multiple regulating means in the future to further improve the anisotropy of the as-obtained nanoclusters. We have given a more comprehensive discussion with reference to alternative approaches and previous reports in the revised manuscript (p. 2, highlight lines). Moreover, we have improved the manuscript from the following aspects based on the comments of reviewers.

(1) We systematically studied the output of the asymmetric transformation process (Supporting Information, p. S5-6). Based on the experimental results, a detailed discussion on the mechanism of the reaction has been added in the revised manuscript (p. 7 and 8, highlight lines) and Supporting Information (Figure S8).

(2) We added a detailed analysis of the exciton-coupled CD profiles with respect to the UV-vis spectra of LC₁-LC₅, and reconducted the CD-ee dependence based on anisotropy at 266, 332, 500, and 610 nm. A clearer statement of significance and applicability of this work has been made. These additional experiments and the corresponding comments have been added in the revised manuscript (Figure 5 and highlight lines in p. 12) and supporting Information (p. S13-15).

The revised manuscript gives a more comprehensive and in-depth understanding of the reaction between achiral metal nanoclusters and chiral ligands.

Reviewer comments, second round

Reviewer #1 (Remarks to the Author):

The authors have responded to my concerns and have revised the manuscript accordingly. This is important work and I suggest publication at this point.

Reviewer #2 (Remarks to the Author):

The revised manuscript has well addressed the comments of mine and I am happy with it that I recommend it publication after addressing the following minor points.

(1) Figure 5 shows only CD spectra to identify if exciton coupling happens or not. Better to show the CD spectra together with the corresponding absorption spectra that the exciton coupling can be well shown. Thus part of the absorption spectra shown in Figure S12 shall be moved to be included in the revised Figure 5.

(2) It is great to be able to obtain the crystals and their structures of the chiral clusters prepared by using the enantiomer pairs of varying ee of the chiral ligands. Even more importantly the crystal was found either R- or S-enantiomer, which means a spontaneous resolution happens during the preparation of the chiral transformation. A linear CD-ee dependence is expected from this observation. Yet experimentally observed CD-ee dependence is nonlinear. From the experimental procedures for preparing their CD-ee dependence it is clear that other chiral clusters than the isolated clusters exist in the total products which likely makes the should-be linear CD-ee change into nonlinear dependence. This shall be clarified. An experiment that may be done is to measure the CD spectrum of the dissolved isolated cluster enantiomer crystal in solution and compare it with that obtained from that prepared from the reaction of achiral cluster with the ligand enantiomer.

Reviewer 1:

No further revision requested.

Reviewer 2:

The revised manuscript has well addressed the comments of mine and I am happy with it that I recommend it publication after addressing the following minor points.

(1) Figure 5 shows only CD spectra to identify if exciton coupling happens or not. Better to show the CD spectra together with the corresponding absorption spectra that the exciton coupling can be well shown. Thus part of the absorption spectra shown in Figure S12 shall be moved to be included in the revised Figure 5.

Response: We thank the reviewer for the advice. We have shown CD spectra together with the corresponding absorption spectra in the revised Figure 5, which is also shown below. The corresponding description has also been added in the text (p. 9, highlight lines) of the revised manuscript.

Figure 5. (a,b) Combined CD-absorption spectra of (Rac) -, (R) -, (S) - L_1 and (Rac) -, (R) -, (S) - LC_1 (Inset: the enlarged view from 400 to 700 nm). (c) Anisotropy factors of LC_1 prepared by different ee values of L_1 , and the CD-ee dependence based on anisotropy.

(2) It is great to be able to obtain the crystals and their structures of the chiral clusters prepared by using the enantiomer pairs of varying ee of the chiral ligands. Even more importantly the crystal was found either R- or S-enantiomer, which means a spontaneous resolution happens during the preparation of the chiral transformation. A linear CD-ee dependence is expected from this observation. Yet experimentally observed CD-ee dependence is nonlinear. From the experimental procedures for preparing their CD-ee dependence it is clear that other chiral clusters than the isolated clusters exist in the total products which likely makes the should-be linear CD-ee change into nonlinear dependence. This shall be clarified. An experiment that may be done is to measure the CD spectrum of the dissolved isolated cluster enantiomer crystal in solution and compare it with that obtained from that prepared from the reaction of achiral cluster with the ligand enantiomer.

Response: We thank the reviewer for pointing out this and the advice of comparing the CD spectrum of the isolated cluster with that of the crude product containing all generated clusters. We have done the experiment and obtained the result, which has been shown below and also in Figure S14 of the revised Supporting Information. From the result, the isolated LC_1 nanocluster showed slightly different CD spectra from that of the crude product which was directly obtained from the reaction of Au_{23} and L_1 . As mentioned by the reviewer, the difference is probably due to other chiral clusters than the isolated cluster existing in the crude product. This might make the should-be linear CD-ee dependence change into nonlinear one. We have clarified this in the text (p. 11, highlight lines) of the revised manuscript.

Figure S14. CD spectra of the isolated (R)- and (S)- LC_1 , and the crude product obtained from the reaction of Au_{23} with (R)- and (S)- L_1 (Inset: the enlarged view from 400 to 700 nm).

Reviewer comments, third round

Reviewer #2 (Remarks to the Author):

I am now happy with the changes made to the second version of the manuscript and therefore recommend publication in its current form.

Yet, I would suggest to delete the sentences start on line 226 "Meanwhile,..." to line 228 "...process".